# Role of Different Variants of Leptin Receptor in Human Adrenal Tumor Types

**DOI:** 10.3390/ijms25168682

**Published:** 2024-08-09

**Authors:** Anna Klimont, Marcin Ruciński, Nadia Sawicka-Gutaj, Marta Szyszka, Małgorzata Blatkiewicz, Tomasz Wierzbicki, Marek Karczewski, Małgorzata Janicka-Jedyńska, Marek Ruchała, Hanna Komarowska

**Affiliations:** 1Department of Endocrinology, Metabolism and Internal Medicine, Poznan University of Medical Sciences, 60-356 Poznan, Poland; anna.klimont@usk.poznan.pl (A.K.); nsawicka@ump.edu.pl (N.S.-G.); mruchala@ump.edu.pl (M.R.); 2Department of Histology and Embryology, Poznan University of Medical Sciences, 60-781 Poznan, Poland; marcinruc@ump.edu.pl (M.R.); mszyszka@ump.edu.pl (M.S.); mblatkiewicz@ump.edu.pl (M.B.); 3Department of General, Endocrinological and Gastroenterological Surgery, Poznan University of Medical Sciences, 60-355 Poznan, Poland; 4Department of General and Transplantation Surgery, Poznan University of Medical Sciences, 60-356 Poznan, Poland; mkar@ump.edu.pl; 5Department of Clinical Pathology, Poznan University of Medical Sciences, 60-356 Poznan, Poland; mjjedynska@ump.edu.pl

**Keywords:** leptin, adrenal carcinoma, ACC, leptin receptor isoforms, steroidogenesis

## Abstract

The aim of the study was to evaluate the diagnostic and prognostic significance of leptin receptor isoforms in adrenal tumors. In a single-center study, 96 patients (19 with adrenal cortical carcinoma and 77 with benign tumors) underwent an adrenalectomy. A total of 14 unaffected adrenal gland tissues from kidney donors were used as controls. Fasting blood samples were collected for laboratory tests, and mRNA expressions of leptin receptor isoforms were assessed by RT-qPCR. The study analyzed correlations between mRNA expressions and clinical data and measured NCI-H295R cell proliferation via a real-time cell analyzer. All adrenal lesions expressed leptin receptor isoforms. Significantly lower LepR1 expression was observed in carcinoma tissues than in adenomas and controls (*p* = 0.016). Expressions of LepR3&LepR6 were correlated with overall survival (*p* = 0.036), while LepR2&LepR4 and LepR5 expressions were inversely related to morning serum cortisol levels (*p* = 0.041). Leptin reduced NCI-H295R cell proliferation (*p* < 0.0001). The study highlights the diagnostic and prognostic significance of leptin receptor isoforms in adrenal tumors. Specifically, LepR1 may serve as a diagnostic marker for carcinomas, while LepR3&LepR6 have potential use as prognostic markers.

## 1. Introduction

Leptin is an adipocytokine that has been studied extensively. This 147-amino acid peptide is a product of the obesity gene (*OB*) and is mostly secreted by adipose tissue [1,2,3]. Together with ghrelin and adropin leptin, it is involved in regulation of body weight and energy balance [4,5,6]. In humans, the plasma leptin concentrations in nonobese subjects are ~10 nmol/L [7]. The plasma leptin concentration correlates with total fat mass and is elevated in obese patients and decreased during fasting [8]. Both leptin and ghrelin may influence adrenal gland function either indirectly through regulation of the hypothalamus–pituitary–adrenal (HPA) axis or directly by affecting adrenal cortical or medullary cells [9,10,11]. Serum leptin concentrations are lower in men than in women [12].

Leptin binds to the specific leptin receptor (LepR), a member of the cytokine 1 receptor family [13], which exists in seven isoforms resulting from alternative splicing during *LEPR* gene expression [14,15,16]. These isoforms are grouped into three categories: long (LepR1), short (LepR2, LepR3, LepR4, LepR5 and LepR6), and secreted or soluble (soluble LepR) forms. All LepR isoforms share an extracellular ligand-binding domain at the N-terminus but differ in the C-terminus. The detailed primary structure of the leptin receptors is shown in Figure 1. Six transcript variants (LepR1, LepR2, LepR3, LepR4, LepR5 and LepR6) possess transmembrane domains, while soluble LepR compress only extracellular domains. Interestingly, there is no mRNA transcript encoding soluble LepR, but it is generated by proteolytic cleavage of membrane-anchored leptin receptors [17]. Leptin binding to all five forms of the short leptin receptor induces Janus kinase (JAK) activation and further phosphorylation of insulin receptor substrates (IRSs), triggering activation of the PI3-K/Akt pathway [18]. The soluble isoform, which lacks an intracellular domain, exerts biological functions. As the primary circulating leptin-binding protein, LepR soluble has been shown to inhibit leptin binding to other LepR isoforms, thereby neutralizing leptin-mediated STAT3 signaling [19].

In addition, the short isoforms of leptin receptors are involved in leptin transport and clearance [12]. In particular, LepR2 and LepR3 are involved in leptin transport across the blood–brain barrier. Due to the ability of leptin to activate different intracellular signaling pathways, depending on the type of receptor isoform and their interactions, leptin can induce different biological consequences. On the other hand, different variants of LepR may be involved in the regulation of leptin levels. Adipose tissue functions as an endocrine gland, secreting adipocytokines such as leptin, adiponectin and visfatin. The links between adipocyte-derived factors and carcinogenesis are being explored in malignancies, including both those associated with obesity [20] and without such an association [21,22,23,24]. Alterations in leptin and leptin receptor protein expression are linked with colorectal cancer, where they are directly associated with proliferation and apoptosis processes [25]. Serum leptin levels in thyroid cancer patients were found to be significantly higher than in those in the control group [26,27,28]. The mRNA expression patterns of leptin receptor isoforms differ between normal prostate and prostate cancer cells [29]. Relatively high leptin expression has been found in many cancers. There are only a few examples of research that have examined the leptin receptor [30].

Adrenal tumors are a heterogeneous group of malignancies with various clinical presentations and prognoses [31,32]. Over 70% of these tumors are adrenal incidentalomas, which are masses found incidentally during diagnostic imaging for unrelated reasons. Adrenal lesions can be classified as benign or malignant and they may be hormonally active or inactive. Malignant tumors include ACC, malignant pheochromocytoma, lymphoma and adrenal metastases. Benign lesions include adenoma, adrenal hyperplasia, pheochromocytoma, myelolipoma, inflammatory tumors or cysts. PHEO and adrenal hyperplasia are hormonally active, and ACC and adenoma may be hormonally active or inactive. Other lesions are hormonally inactive. Although many adrenal masses are hormonally inactive and appear benign, it is crucial for clinicians to evaluate whether they are functional, nonfunctional, or malignant [33]. Between 2 and 9% of ACA produce glucocorticoids. The incidence of aldosterone-producing tumours is similar. Rarely, ACA may also produce androgens or estrogens. Tumors of the adrenal medulla are called pheochromocytomas (PHEO) [34]. Myelolipomas are a type of adrenal tumour, and they consist mainly of mature adipocytes and are present in 8% of all cases. In our investigation, we focused on ACC, ACA, PHEO, adrenal hyperplasia and myelolipoma.

Adrenal cortical carcinoma (ACC) is a rare and highly aggressive tumour, with a prevalence of 0.5–2 cases per million per year [35]. ACC is more common in women [36]. The clinical presentation of ACC depends on the tumour’s hormonal activity. About 50% of patients with hormonally active tumours manifest symptoms of hypercortisolemia, while 40% present with androgenization and, very rarely, hyperaldosteronism [37]. Patients with nonfunctioning ACC present symptoms of the tumour mass or distant metastases. The histopathological diagnosis of ACC can be challenging. Typically, the modified Weiss criteria are used, although new systems are continually evolving [38]. The Ki67 protein is a cellular marker for proliferation. Ki67 is widely utilized as a prognostic marker in clinical settings due to its specificity for proliferating cells and its detectability across all phases of the cell cycle. Ki67 expression has been documented as a prognostic indicator, aiding in the prediction of cancer survival rates and the probability of recurrence. This marker has been studied across various cancer types, including non-Hodgkin lymphomas, multiple myeloma, soft tissue sarcomas, prostate cancer, and breast cancer [39]. A Ki67 labelling index above 10% indicates aggressive ACC growth and is associated with a poor prognosis [40]. Both endocrine and paracrine mechanisms may be involved in adrenal tumour pathophysiology. Over the last decade, there has been an extensive search for new diagnostic and prognostic markers of ACC. Some of the candidates that have been considered include leptin, ghrelin, adiponectin, adropin and visfatin [24,41,42,43,44].

The literature currently available does not provide any data evaluating the expression patterns of different leptin receptor isoforms in adrenal tumours. The aim of our study was to analyze the mRNA expression of various isoforms of leptin receptors in a group of adrenal tumours and healthy adrenals. The results were then correlated with clinical data from patients who underwent surgery in a single clinical centre. Due to the rarity of these conditions, our study group is relatively large and provides a meaningful sample for analysis. Obtaining a sufficient number of cases, especially those suitable for surgical intervention, is a challenge in clinical research. Our research aims to fill this gap by exploring the potential of leptin receptor isoforms as diagnostic and prognostic markers, which could ultimately lead to faster and more accurate clinical decision-making.

The effects of human leptin treatment on cell proliferation were assessed using the NCI-H295R adrenocortical cancer cell line model. Our paper is the first to investigate the potential pathological involvement of leptin receptor isoforms in adrenal tumors’ pathogenesis.

## 2. Results

### 2.1. Leptin Receptor Isoforms Are Expressed in All Types of Adrenal Lesions and Are Significantly Downregulated in Adrenocortical Carcinoma

We found the expression of examined isoforms in all types of adrenal lesions, as shown in Figure 2. There were statistical differences in expression levels of LepR1, LepR2&LepR4, and LepR3&LepR6 within the studied groups (*p* < 0.05). Expression of LepR v1 was significantly lower in carcinoma tissues compared to adrenocortical adenoma and controls (*p* = 0.016).

### 2.2. Expression of LepR3&LepR6 mRNA Is Correlated with Overall Survival, Whereas Expression of LepR5 Is Negatively Correlated with Overall Survival

Furthermore, we conducted an analysis on whether there is a relationship between the levels of mRNA expression in LepR isoforms, the clinical outcomes of patients and the histopathological characteristics of studied tissues. We identified a correlation between the mRNA expression of LepR3&LepR6 and overall survival (R = 0.58, *p* = 0.036), as seen in Figure 3A. Additionally, correlations were noted between the mRNA expression of LepR5 and Ki67 (R = 0.65, *p* = 0.0068), as seen in Figure 3B.

### 2.3. Expression of LepR2&LepR4 and LepR5 Is Negatively Correlated with Morning Serum Cortisol Concentration

There were negative correlations between *mRNA* expression of LepR2&LepR4 and LepR5 and morning serum cortisol concentrations (R = −0.29 *p* = 0.04; R = −0.29 *p* = 0.041).

### 2.4. Effect of Leptin on Proliferative Human Adrenocortical Cell Line

The analysis of the impact of human leptin peptide on the proliferative abilities of the human adrenocortical cell line revealed a significant decrease in the normalized cell index—NCI (Figure 4). We indicate that administering both leptin concentrations in NCI-H295R cell lines similarly reduces cell proliferation (*p* < 0.0001). Moreover, the impact of the tested substances on the growth rate remained constant across the cultivation time. Furthermore, we performed a comprehensive analysis at 24, 48, and 72 h following the introduction of leptin.

## 3. Discussion

Indications for adrenalectomy are limited to lesions with suspicion of malignancy and those that are hormonally active. Our study group should be considered significant, especially as ACC is very rare. Furthermore, the diagnosis of ACC is still difficult [45], and its treatment is a challenge because many patients do not respond to available therapies. Therefore, new markers and new treatment options are still being sought [46]. There are studies confirming the role of leptin in cancerogenesis. We decided to assess expression of leptin receptors in different types of adrenal tumors and the influence of leptin on ACC cell lines. To the best of our knowledge, a singular study has delved into leptin receptor mRNA expression in adrenal gland tumors [47]. The majority of research has traditionally assessed leptin and receptor expression at the protein level using immunohistochemistry methods. Our study, however, stands as a pioneering effort to evaluate the expression of distinct receptor isoforms at the mRNA level in adrenal gland lesions, comparing them to healthy controls.

In our investigation, we identified the mRNA expression of various LepR isoforms across all analyzed types of adrenal tumors, providing evidence that supports leptin’s potential involvement in adrenal tumorigenesis.

Currently, there are six known splicing variants of the leptin receptor. Sequence analysis of the variants deposited in the NCBI database indicates significant changes in the nomenclature of individual variants, as shown in Figure 5.

The first paper regarding leptin receptors mRNA expression in adrenal glands tumors was published in 1999 by Glasow et al. [47]. Their study aimed to scrutinize potential variations in LepR mRNA expression within human adrenal tumors. The authors conducted experiments utilizing cDNA from eight adrenal tumors and three distinct RNA isolations from the ACC cell line (NCI H295R). Their findings revealed mRNA expression of the full-length LepR and isoforms B219.1 and 3 in each of the investigated ACC and adenomas. Notably, the Ob-R B219.2 isoform was expressed to a lesser extent and detected only in three out of eight samples. The ACC NCI-H295 cell line expressed all forms of LepR, excluding the B219.2 isoform. However, the researchers did not compare these results with receptor expression in normal adrenal tissue. In terms of nomenclature, B219.1 is equivalent to LepR4, B219.2 corresponds to LepR5, while B219.3 represents LepR6 [29]. Szyszka et al., based on current leptin receptor gene structure suggested that researchers determine the total expression of LepR3 and LepR6, since it might be impossible to design primaries only for LepR6.

We showed lower expression on a mRNA level of the so-called long leptin receptor (LepR1) in ACC compared to controls and ACA. This suggests that reduced expression of the LepR1 contributes to the development of adrenocortical carcinoma.

We identified an interesting pattern of isoform expression with potential prognostic significance. A correlation was found between the mRNA expression of LepR3&LepR6 and overall survival. This suggests a potentially protective role for this receptor variant. On the other hand, for LepR5, a strong correlation was observed between its mRNA expression and Ki67. This receptor variant might serve as a candidate for poor prognosis markers. However, both findings necessitate further investigation on larger patient cohorts.

Additionally, we observed a negative correlation between mRNA expression of LepR2&LepR4 and LepR5 and morning serum cortisol concentration. LepR2&LepR4 expression was significantly downregulated in adenomas, pheochromocytomas and adrenal gland hyperplasia compared to the control group and myelolipoma. Our findings align with earlier studies that reported leptin’s potential to influence the hypothalamus–pituitary–adrenal (HPA) response to stress and starvation in animal models [48,49]. Notably, LepR2 mediates the transport of leptin through the blood–brain barrier [50], and these isoforms might play a role in impairing the HPA response.

In our study, we observed a significant upregulation of LepR3&LepR6 expression in ACA and myelolipoma compared to the control group and ACC. This underscores the potential relevance of these variants in metabolic observations related to leptin and warrants further investigation.

Moreover, our studies on the effect of leptin on proliferation of the human adrenocortical cell line showed that leptin at a physiological concentration inhibits proliferation. We indicated that administering both leptin concentrations 10 nmol/L and 0.1 nmol/L in NCI-H295R cell lines significantly reduces cell proliferation. The impact of the tested substances on the NCI remained constant across the cultivation time. Furthermore, we performed a comprehensive analysis at 24, 48, and 72 h following the introduction of leptin. Previous studies regarding the effects of human leptin on proliferation described conflicting results. In the only paper on the adrenal glands in 1999, Glasow and coworkers attempted to assess the proliferative effect of the physiological concentration of leptin in NCI-H295R cell lines [47]. They found that, although the leptin receptor is expressed on both the mRNA and protein level in adrenal tumors, leptin does not regulate the proliferation of neoplasms. We know that ACCs comprise a very heterogeneous group, and this may account for differences in our studies. Based on our research, we can assume that leptin inhibits the proliferation of adrenocortical carcinoma cells.

As for the impact of leptin on various cancer cells, both inhibitory and stimulatory effects have been observed. For instance, in mouse mammary cancer cell lines treated with leptin, researchers demonstrated an induction of proliferation, attributing it to the crosstalk between Notch-IL-1-Leptin. This interplay of leptin, IL-1 and Notch upregulated ligands, receptors and relevant genes, thereby enhancing the proliferation and migration of cancer cells [51]. Similar results were described in relation to pancreatic cancer [52]. The same crosstalk of IL-1 and leptin was associated with greater invasiveness and chemoresistance in endometrial cancer [20]. Conversely, Weichhaus et al. found that supraphysiological concentrations of leptin inhibited the proliferation of breast cancer cells by suppressing MAPK signaling [53]. In a study on papillary thyroid carcinoma cells, researchers reported that supraphysiological doses of leptin stimulated both proliferation and migration in vitro [54]. Notably, the concentrations of leptin used in our research were within the physiological range. Liu et al. explored the effects of leptin on cytotrophoblast cells, finding that, while the proliferation of cytotrophoblast was inhibited after treatment with different leptin concentrations, a supraphysiological concentration significantly enhanced invasiveness and migration compared to the controls [55].

Knowledge of the molecular mechanisms and signaling pathways involved in adrenocortical tumorigenesis is still insufficient. The most common alterations are constitutive activation of the Wnt/b-catenin signaling pathway and activation of the PI3K/AKT/mTOR cascade and RAS-MAPK pathway by IGF2 [56,57]. Other signaling pathways may also play a role in ACC tumorigenesis [23].

In a previous study, we evaluated the effect of adropin on adrenal cancer cell lines (HAC15) and its receptor (GPR19) expression in adrenal tumor samples [42]. We found that GPR19 expression is elevated in cancerous tissues compared to healthy adrenal glands and that adropin significantly stimulated the proliferation of HAC15 cells. This relationship between adropin and leptin is intriguing, as both proteins act antagonistically during energy homeostasis and the proliferation of adrenal cancer cells. Notably, both receptors, when combined with their ligands, can activate the MAPK/ERK1/2 signaling pathway, which appears to be crucial in the development of many cancers. It is possible that leptin acting through LepR1 via the MAPK-dependent signaling pathway inhibits cell proliferation in adrenal tumors.

To the best of our knowledge, our study is the first to suggest the potential of leptin treatment to inhibit cell proliferation in adrenocortical cancer cells. One potential explanation is that leptin might downregulate the active isoform of the leptin receptor, LepR1, in ACC cells, thereby impairing the activation of intracellular pathways, including Notch-IL-1 and leptin crosstalk [51,52]. Another proposed explanation for the inhibited proliferation comes from Thompson et al., who evaluated the effects of leptin on hepatocellular carcinoma (HCC) cells. They found that physiological concentrations of leptin inhibit HCC cell growth in vitro, suggesting a mechanism via a p38-MAPK-dependent signaling pathway [58].

In summary, the impact of leptin on cancer cell proliferation appears to vary based on the cancer type and the concentration of leptin used in experiments. It is crucial to note that, in our study, we did not evaluate total leptin receptor expression; instead, we categorized it into subtypes. It is plausible that, in carcinogenesis, receptor isoforms other than LepR1 might play a role and modify the final effect of leptin on LepR1.

## 4. Materials and Methods

### 4.1. Study Cohort and Strategy

The study included 96 patients diagnosed with adrenal lesions, comprising 19 patients with ACC and 77 patients with benign tumors (adenoma, adrenal hyperplasia, myelolipoma and pheochromocytoma). All patients were hospitalized in the Department of Endocrinology and Internal Medicine at the Poznan University of Medical Sciences between 2011 and 2022. The indications for adrenalectomy were suspicion of malignancy or hormonal activity of the tumour. The control group comprised 14 unchanged adrenal gland tissues obtained from kidney donors.

The adrenocarcinoma was diagnosed histopathologically using Weiss criteria [35]. The tumor size and tumor stage at diagnosis were classified according to the European Network for the Study of Adrenal Tumors’ classification [59]. The study protocol received approval from the Bioethics Committee of Poznan University of Medical Sciences (decision No. 31/22) [60].

Fasting blood samples were collected from each patient, and a series of laboratory tests were conducted. Levels of fasting plasma glucose, thyroid-stimulating hormone (TSH), free thyroxine (fT4), total cholesterol, high-density lipoprotein (HDL), low-density lipoprotein (LDL), triglycerides, morning cortisol, evening cortisol, cortisol in post-dexamethasone suppression test, adrenocorticotropic hormone (ACTH), dehydroepiandrosterone sulfate (DHEA-S), testosterone, sex hormone-binding globulin (SHBG), androstenedione, sodium, potassium and 17-Hydroxyprogesterone (17-OHP) were evaluated. Clinical characteristics of the patients with adrenal gland tumors are shown in Table 1.

Following surgical removal, sections of the pathologically altered adrenal glands (approximately 0.5 cm^3^ in size) were promptly immersed in RNA later, then in Tissue Storage Reagent (Sigma, Kawasaki, Japan), and then stored at −70 °C until RNA isolation was performed.

### 4.2. RNA Isolation

Total RNA was extracted from human adrenal gland samples using TRI reagent (Sigma–Aldrich, Poznan, Polska) and a Universal RNA Purification Kit (#E3599-02, EURx, Gdansk, Polska) according to the manufacturer’s protocol. The concentration of total RNA was determined by measuring the optical density at 260 nm, and its purity was estimated by a 260/280 nm absorption ratio (higher than 1.8) (NanoDrop spectrophotometer, Thermo Fisher Scientific, Waltham, MA, USA). The extracted RNA samples were stored at −80 °C for further quantitative polymerase chain reaction (qPCR) analysis.

### 4.3. Reverse Transcription (RT)

Reverse transcription was performed using the reverse transcriptase from the Transcriptor First Strand cDNA Synthesis kit (Roche Diagnostics Corp., Indianapolis, IN, USA) with Oligo(dT) primers at a temperature of 55 °C for 40 min (Thermocycler UNO II, Biometra GmbH, Göttingen, Germany). Each RT reaction used 1 µg of total RNA and was carried out with a standard final volume of 20 µL. Subsequently, each cDNA-containing sample was diluted with 100 µL of RNase-free water.

### 4.4. QPCR

The primers used in this study were designed and previously published [61]. The primers were designed by Primer 3 software (Whitehead Institute for Biomedical Research, Cambridge, MA, USA) and purchased from the Laboratory of DNA Sequencing and Oligonucleotide Synthesis, Institute of Biochemistry and Biophysics, Polish Academy of Sciences, Warsaw. Table 2 and Figure 6 present the characteristics of the primers and their binding sites to exons specific for individual isoforms of the leptin receptor. The study used various combinations of primer pairs, each marked with corresponding numbers on Figure 2, as individual primer pairs do not allow for specific amplification of each isoform. Primer pair 1 amplifies only isoform Lepr1(LepR1), while pair 2 amplifies isoforms 2 and 4 (LepR2&LepR4). Pair 3 amplifies isoforms 3 and 6 (LepR3&LepR6), and pair 4 specifically amplifies isoform 5 (LepR5). The qPCR was carried out using the Thermocyckler (CFX96, Bio-Rad, Hercules, CA, USA) with a SYBR Green detection system. Each reaction mixture (10 µL) contained 2 µL of template cDNA, 0.5 µM of each specific primer, an SYBR green kit from Bio-Rad (SsoAdvanced Universal SYBR Green cat. #1725271, Hercules, CA, USA) and Nuclease-free water. The thermal cycling conditions for running the PCR reaction plate were as follows: UNG incubation (2 min at 50 °C), AmpliTag Gold^®^ and UP Enzyme activation (10 min at 95 °C), followed by 40 cycles of denaturation (15 s at 95 °C) and annealing (1 min at 60 °C). All samples were amplified in duplicate. The specificity of the reaction products was verified by the determining of melting points at a transition rate of 0.1 °C/s. The relative expression of the target genes was calculated using a 2^−ΔΔCt^ quantification method. For expression normalization, two different housekeeping genes, HPRT and B2M, were used as reference genes.

### 4.5. Adrenocortical Carcinoma Cell Line and Treatment

The NCI-H295R human cell line (#CRL-2128, ATCC, Manassas, VA, USA) was obtained from the American Type Culture Collection (ATCC, VA, USA) and cultured in a defined medium according to the manufacturer’s protocol. Briefly, the DMEM:F-12 medium (#30-2006, ATCC, VA, USA) contained 1% Penicillin/Streptomycin (Merck Millipore, Darmstadt, Germany), 1% ITS + Premix (#354352, Corning, Corning, NY, USA), and 2.5% Nu-Serum (#355100, Corning, NY, USA). The cell line was cultured in 75 cm^2^ flasks in a humidified incubator at 37 °C in 5% CO_2_. The medium was renewed three times a week. After 24 h of cultivation, the cells were treated with leptin peptide (#300-27, PreproTech, Cranbury, NJ, USA) in two concentrations (1 × 10^−8^ and 1 × 10^−10^).

### 4.6. Cell Proliferation by RTC

The proliferation of NCI-H295R cell was assessed using a real-time cell analyzer (RTCA, Roche Applied Science, Penzberg, Germany). For the RTCA analysis, cells were cultured in 16-hole slide plates (E-Plate 16) at a concentration of 25,000 cells per well in 200 µL of medium. Changes in electrical impedance were measured by sensors placed at the bottom of the chamber, enabling real-time data collection every 15 min during the experiment. Continuous monitoring of cell division was enabled by this method. The RTCA assay’s crucial parameter is the cell index (CI), which reflects fluctuations in electrical impedance that correspond to the growth or decline rate of the cultivated cells. To standardize CI values, a normalized cell index (NCI) was calculated using the formula NCI = CI time point/CI at substance delivery. The ggplot2 package was used to visualize the mean NCI values along with standard deviation (SD) [61].

### 4.7. Statistical Analysis

Normalized expression values for all analyzed primer pairs were presented as medians with an interquartile range (IQR) using the ggplot2 package. Statistical differences were determined by the Kruskal–Wallis test followed by Dunn’s post hoc test. A *p*-value of <0.05 was considered statistically significant. Groups sharing the same letter are not significantly different according to the Dunn’s test. Correlation analysis was performed using the Pearson correlation test [62].

## 5. Conclusions

In conclusion, we have identified distinct patterns of leptin receptor isoform expression in adrenal tumors, underscoring leptin’s significant role in oncogenesis. Our findings reveal that LepR1 expression is notably lower in adrenocortical carcinoma (ACC) compared to other adrenal lesions and healthy controls, suggesting its potential as a diagnostic marker. Moreover, the expression of LepR3 and LepR6 has been correlated with overall survival, indicating a possible protective role, while LepR5 expression is associated with poor prognosis, as evidenced by its correlation with Ki67 levels. These observations highlight the multifaceted role of leptin in tumor biology, including its impact on cellular proliferation, migration, and the modulation of signaling pathways. This study also provides the first evidence of leptin’s ability to inhibit cell proliferation in ACC, pointing to its potential therapeutic implications. Further research is warranted to explore the clinical utility of these findings in the diagnosis, monitoring and prognosis of adrenal tumors.

## Figures and Tables

**Figure 1 ijms-25-08682-f001:**
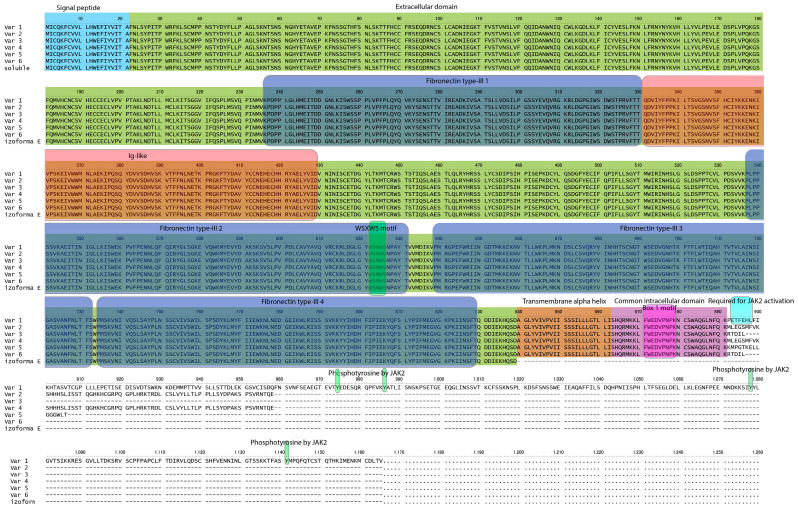
Comparison of the amino acid sequences of leptin receptors with indication of individual functional domains and differences between leptin receptor variants. Data based on the Uniprot database.

**Figure 2 ijms-25-08682-f002:**
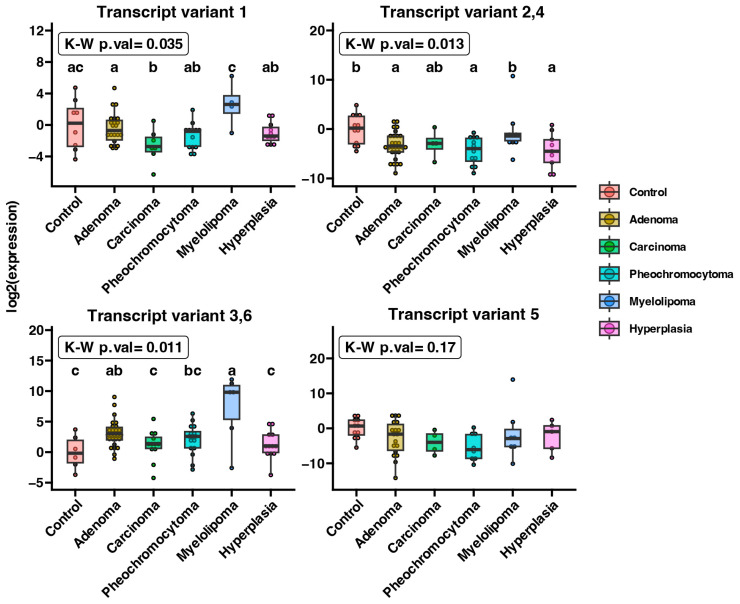
Expression of LepR transcript variants examined in healthy adrenal glands (control) and across five clinical types of adrenal tumors (adenoma, carcinoma, pheochromocytoma, myelolipoma, hyperplasia). Box plots were used to display the median, interquartile range, minimum and maximum values. Normalized expression levels for individual patients were represented as dots. Statistical differences were assessed using the Kruskal–Wallis test followed by Dunn’s post hoc test. Groups that share the same letter are not considered significantly different based on Dunn’s test.

**Figure 3 ijms-25-08682-f003:**
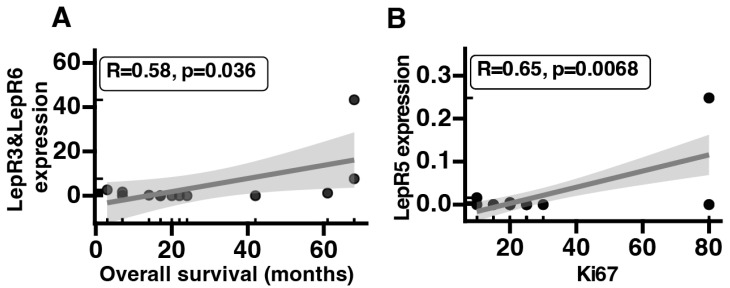
Correlation of LepR3&LepR6 expression levels with overall survival (**A**). Correlation of LepR5 expression with Ki67 (**B**).

**Figure 4 ijms-25-08682-f004:**
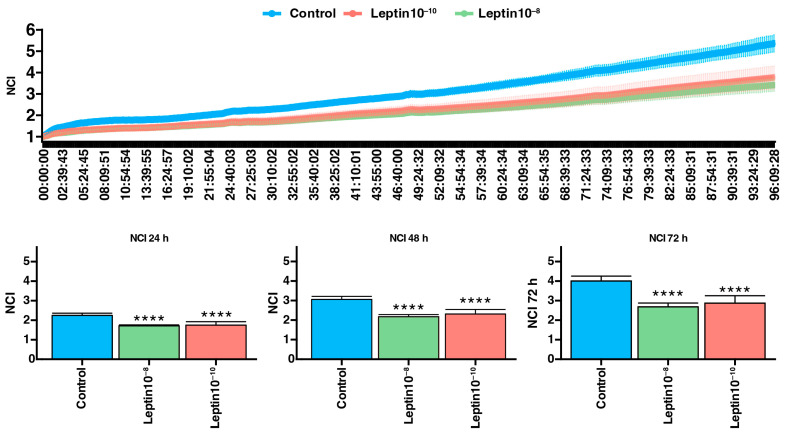
The H295R cell line’s normalized cell index (NCI) monitored for 96 h after treatment with human leptin peptide. The results showed a consistent trend of reduced proliferation post-administration (**** *p* < 0.0001). The RTCA chart illustrates the mean NCI ± SD, with the control group represented by the blue line, leptin 10^−8^ treatment by the green line and leptin 10^−10^ treatment by the red line. The NCI values at selected time points (24, 48, 72 h) following peptide administration are presented below.

**Figure 5 ijms-25-08682-f005:**
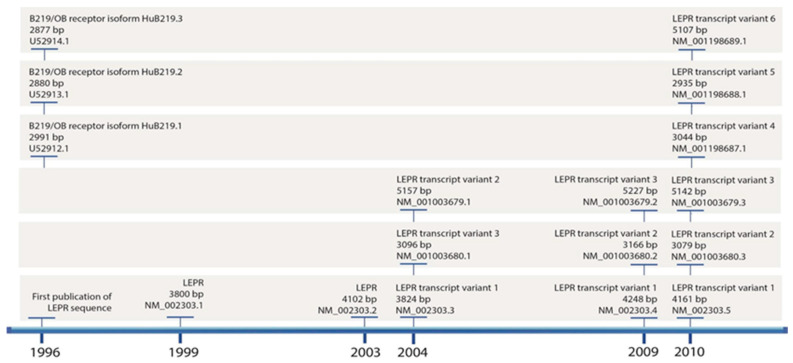
Sequence of changes in the nomenclature of individual leptin receptor transcript variants along with their length and ID numbers, based on NCBI data. The *x*-axis of the graph represents the time range of entries appearing in the NCBI database.

**Figure 6 ijms-25-08682-f006:**
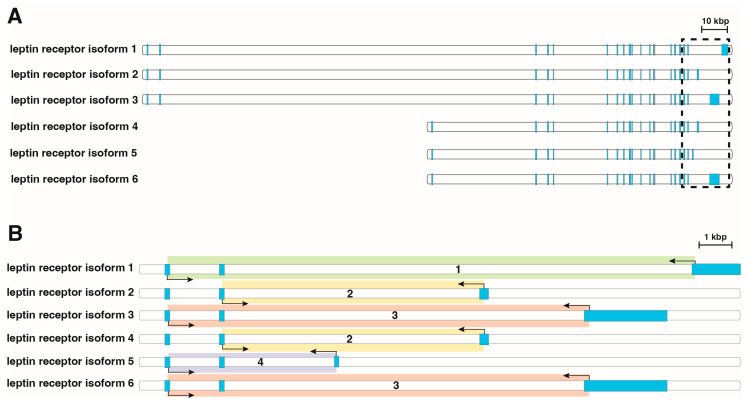
Structure of the human splicing variants of the leptin receptor gene (**A**). The 3’ region, to which the primers bind, is framed with a dashed line. The detailed structure of this region is shown in panel (**B**). The exons are represented by vertical blue blocks, where the width of each block is equal to the length of the individual exon. The primer binding sites for all distinguishable transcription variants of the leptin receptor gene are indicated by arrows. The resulting amplicons are highlighted with separate colors. The overall structure of the leptin receptor gene isoforms, with consideration of the location of individual exons, was analyzed using DNAsis Max ver. 2.5 (Hitachi Ltd., Tokyo, Japan). The primer binding site positions were marked manually using Adobe Illustrator ver. 28.4 (Adobe Inc., San Jose, CA, USA). Each pair of primers is assigned a specific number that corresponds to the numbering in Table 2.

**Table 1 ijms-25-08682-t001:** Clinical characteristics of the patients with adrenal gland tumors. Numerical values are presented as median, with first and third quartiles in brackets.

Variable	ACC, N = 19	Adenoma, N = 35	Hyperplasia, N = 14	Myelolipoma, N = 10	Pheochromocytoma, N = 18
Tumor diameter (mm)	115 (74, 155)	29 (16, 43)	38 (24, 57)	65 (55, 77)	50 (36, 60)
Ki67%	20 (10, 30)	NA (NA, NA)	NA (NA, NA)	NA (NA, NA)	NA (NA, NA)
Overall Survival (months)	18 (7, 61)	NA (NA, NA)	NA (NA, NA)	NA (NA, NA)	NA (NA, NA)
FPG (mg/dL)	94 (86, 113)	96 (89, 107)	135 (94, 202)	91 (86, 96)	107 (98, 130)
TSH (µU/L)	1.56 (1.06, 1.83)	1.35 (0.74, 2.71)	1.17 (0.98, 1.34)	2.01 (1.80, 2.02)	1.67 (1.66, 1.75)
Ft4 (pmol/L)	16.50 (13.22, 17.30)	16.30 (14.60, 18.30)	16.06 (14.93, 16.37)	20.54 (16.93, 21.25)	15.79 (15.69, 16.24)
Total cholesterol (mg/dL)	178 (143, 220)	212 (170, 241)	175 (164, 216)	152 (139, 161)	219 (177, 248)
HDL (mg/dL)	47 (37, 62)	56 (50, 73)	134 (86, 318)	58 (50, 109)	88 (57, 103)
LDL (mg/dL)	108 (73, 128)	108 (83, 143)	61 (47, 71)	64 (43, 90)	94 (61, 145)
TAG (mg/dL)	132 (93, 159)	140 (103, 187)	106 (83, 185)	107 (93, 107)	120 (103, 154)
Morning cortisol (nmol/L)	427 (340, 633)	500 (428, 600)	719 (444, 782)	367 (237, 587)	482 (402, 630)
Evening cortisol (nml/L)	315 (150, 534)	160 (91, 296)	397 (171, 753)	193 (146, 391)	138 (69, 177)
Post-DST (nmol/L)	452 (225, 611)	420 (125, 639)	162 (162, 162)	NA (NA, NA)	1314 (1314, 1314)
ACTH (pg/mL)	26 (2, 39)	14 (6, 31)	10 (6, 21)	36 (22, 122)	48 (30, 79)
DHEAS (µg/dL)	164 (52, 534)	70 (31, 118)	108 (58, 125)	100 (43, 155)	125 (83, 278)
Testosterone (nmol/L)	4 (1, 12)	1 (0, 4)	4 (2, 6)	17 (14, 18)	17 (9, 19)
SHBG (nmol/L)	36 (28, 68)	50 (40, 65)	100 (100, 100)	NA (NA, NA)	49 (33, 76)
Androstendione (ng/mL)	2.89 (1.31, 8.63)	1.29 (0.43, 1.44)	3.83 (3.79, 3.86)	0.75 (0.75, 0.75)	3.50 (3.04, 4.07)
17-OHP (ng/mL)	2.33 (0.72, 3.27)	1.07 (0.76, 1.61)	0.82 (0.82, 0.82)	20.00 (20.00, 20.00)	5.92 (5.92, 5.92)
Na (mmol/L)	141.00 (139.00, 143.50)	141.00 (140.00, 143.00)	142.00 (140.50, 142.75)	141.00 (141.00, 141.00)	141.00 (140.00, 142.00)
K (mmol/L)	4.37 (4.02, 4.63)	4.37 (4.06, 4.58)	4.20 (3.58, 4.43)	4.56 (4.33, 4.82)	4.57 (4.21, 4.69)

**Table 2 ijms-25-08682-t002:** PCR Primer Characteristics. S—sense primer, A—antisense primer.

Primers Number	Transcript Variant	Primer	Primer Sequence (5′-3′)	Position	PCR Product Size (bp)
1	Transcript variant 1 of leptin receptor	S	CAGAGTGATGCAGGTTTATATG	2694–2715	208
		A	CTGATGCTGTATGCTTGATAA	2901–2881	
2	Transcript variant 2, 4 of leptin receptor	S	GCCAGTAATTATTTCCTCTTCC	2723–2744	198
		A	CCCTGGGTACTTGAGATTAG	2920–2901	
3	Transcript variant 3, 6 of leptin receptor	S	TATGTAATTGTGCCAGTAA	2712–2730	200
		A	ACATTGGGTTCATCTGTAGTG	2911–2891	
4	Transcript variant 5 of leptin receptor	S	TCCCCATTGAGAAGTACCAGT	2555–2575	300
		A	CACCCAGTAGTTCCTTTGTGC	2854–2834	
-	HPRT	S	GCCATCACATTGTAGCCCTC	343–362	172
		A	ACTTTTATGTCCCCTGTTGACT	514–493	
-	Beta -2-Microglobulin	S	ATGATGCTGCTTACATGTCTC	261–280	170
		A	AAAAGTGGAGCATTCAGACT	410–430	

## Data Availability

All of the data discussed in this work, if not already included in the manuscript, are available from the corresponding author on reasonable request.

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
