# Peer review of "Role of Different Variants of Leptin Receptor in Human Adrenal Tumor Types"

_ijms, 2024, doi:10.3390/ijms25168682_

Round 1
Reviewer 1 Report
Comments and Suggestions for Authors
The present paper by Anna Klimont et al. deals with the possible involvement of Leptin receptor isoforms in various human adrenal tumor types.
General comment :
The study implies a limited number of patients in each category of adrenal tumors, and the rationale for focusing on Leptin receptors is not obvious to the non-specialist. The argument that it is the first study relating Leptin receptor isoforms with adrenal tumors is insufficient and must be addressed more deeply to be convincing.
Specific comments :
· Title’s end : « tumor types » would be more adequate than « tumors ».
· In the Introduction, it is unclear before reading the rest of the paper that « LepRn » equals « LepR var n ». This should be quoted earlier in the Introduction.
· lines 48-49 : « soluble LepR » instead of « LepR soluble »
· Lines 95-96 : Ki-67 as a marker should be introduced more clearly. Is it specific to adrenal tumors or present in all tumors? Its significance in the frame of the study should be more clearly established.
· Figure 2 : it would be easier to compare the data visually if the ordinate scales were the same ; all from -20 to +20, for example.
· Figures 3A and 3B are very difficult to understand: The ordinate titles should be shown vertically on the left of figures 3A and 3B. The units for the abscissas are not shown. The R2 value is low, and the shown regression is very unconvincing.
· Like Figure 3, figure 4 is difficult to read. The pastel colors and figure format do not allow precise data examination. Why only two very different Leptin concentrations (10-10 and 10-8M) ? Why not many more concentrations and endpoint readings ?
· In the lower part of Figure 4, the three panels should all be presented with the same scale from 0 to 5 to allow visual comparison between them.
· In Table 1, it should be indicated that the three figures are the mean plus the lowest and uppest values in brackets for the group considered (If this is indeed the case). But, these pieces of information are not used for discussion.
· Line 329 : correct « transition ».
· In table 2, add the meaning of S and A in the Primer column.
· It would be easier to read if the Conclusions were just after the Discussion.
Author Response
General comment :
The study implies a limited number of patients in each category of adrenal tumors, and the rationale for focusing on Leptin receptors is not obvious to the non-specialist. The argument that it is the first study relating Leptin receptor isoforms with adrenal tumors is insufficient and must be addressed more deeply to be convincing.
Thank you for your thoughtful comment. We acknowledge the concern regarding the sample size of each category of adrenal tumors. However, due to the rarity of these conditions, our study group is relatively large and provides a meaningful sample for analysis. Adrenocortical carcinoma (ACC) is uncommon, and obtaining a sufficient number of cases, especially those suitable for surgical intervention, is a challenge in clinical research. The focus on leptin receptors in this study is motivated by the urgent need for new diagnostic tools that can expedite the identification and characterization of these tumors. We have added an explanatory paragraphs to the manuscript both in introduction and discussion addressing your comment.
Specific comments :
- Title’s end : « tumor types » would be more adequate than « tumors ».
We agree with this comment therefore we have changed the title.
- In the Introduction, it is unclear before reading the rest of the paper that « LepRn » equals « LepR var n ». This should be quoted earlier in the Introduction.
We agree with this comment regarding the clarity of the terminology used in the Introduction. To address this, we have unified the nomenclature throughout the manuscript.
- lines 48-49 : « soluble LepR » instead of « LepR soluble »
We agree with this comment, therefore we have changed text according to the suggestion (line 58).
- Lines 95-96 : Ki-67 as a marker should be introduced more clearly. Is it specific to adrenal tumors or present in all tumors? Its significance in the frame of the study should be more clearly established.
Thank you for your suggestion. We have addressed your comment by providing a clearer introduction of Ki-67 as a marker. We have clarified that Ki-67 is not specific to adrenal tumors but is a widely recognized marker for proliferating cells present in various tumor types on page 3 of the manuscript, lines 106-114.
- Figure 2 : it would be easier to compare the data visually if the ordinate scales were the same ; all from -20 to +20, for example.
Thank you for your suggestions. However, our aim was to examine the changes in expression between the different tumour types and the control group in relation to individual leptin receptor isoforms. Displaying the expressions on the same scale range would have made low-value changes poorly visible, so we left a separate range on the y-axis for each isoform.
- Figures 3A and 3B are very difficult to understand: The ordinate titles should be shown vertically on the left of figures 3A and 3B. The units for the abscissas are not shown. The R2 value is low, and the shown regression is very unconvincing.
The figure was corrected in current version of manuscript
- Like Figure 3, figure 4 is difficult to read. The pastel colors and figure format do not allow precise data examination. Why only two very different Leptin concentrations (10-10 and 10-8M) ? Why not many more concentrations and endpoint readings ?
The leptin concentrations in the presented study were based on previous literature data and our own previously published studies. We only excluded the highest concentration (10^-6), as it is far from physiological values.
The colors have been corrected in the current version of the figure.
- In the lower part of Figure 4, the three panels should all be presented with the same scale from 0 to 5 to allow visual comparison between them.
The scale has been corrected in accordance with the reviewer's suggestions
- In Table 1, it should be indicated that the three figures are the mean plus the lowest and uppest values in brackets for the group considered (If this is indeed the case). But, these pieces of information are not used for discussion.
This information has been added in the current version
- Line 329 : correct « transition ».
We have corrected this.
- In table 2, add the meaning of S and A in the Primer column.
We have added an explanation of the abbreviations to the table description.
- It would be easier to read if the Conclusions were just after the Discussion.
We agree that placing the Conclusions immediately after the Discussion could improve the readability of the manuscript. However, the current structure, with the Conclusions following the Materials and Methods section, adheres to the formatting guidelines prescribed by the journal.
We are grateful for your numerous accurate and meticulous comments. Implementing your suggestions has improved the quality of our publication. Thank you for your valuable contributions.

Reviewer 2 Report
Comments and Suggestions for Authors
Dear Editors,
You can find the references for my comments on the manuscript in the annotated PDF file that I have attached.
Regards,
Abstract
1. Line 20: I don't think it's very correct or advisable for a sentence to start with a number. Please modify it.
2. Lines 29-30: I think these conclusions should be formulated more specifically, especially since the authors obtained results that are quite clear. LepR1 can be a diagnostic marker for carcinomas, LepR3, and LepR4 can be used as prognostic markers.
3. Line 25: Instead of using the complicated notation "LepRv1" the authors might use simply "LepR1" following the notation model of the other receptor variants.
1. Introduction
4. Lines 44-48: See observation 3) above.
5. Lines 60-65: The authors do not use any citations in this text. Indeed, the text contains textbook information but 1-2 citations would be welcome, especially for readers who are not very familiar with the subject and are doing preliminary research.
6. Line 79: Before listing the incidence and some clinical and pathophysiological characteristics of these tumors, a brief listing or even a brief classification of these would be useful. The authors should specify which pathological entities they will focus on in their investigation.
7. Lines 95-96: Use the same notation for the expression of the Ki67 protein (either Ki67 or Ki-67) throughout the manuscript.
8. Line 108: The term "cancer" is quite imprecise, maybe the term "tumor" should be used (possibly enumerate).
4. Materials and Methods:
9. Line 265: A subsection, "Study Cohort and Strategy" (or something similar) for the text between the labels on lines 265-284 needs to be introduced considering there are major sections further below and also since this section includes the study group, the study stages, and the data collection methodology
10. Line 289, Table 1: What values for this variable were recorded? Maximum values or an average, or were multiple diameters measured in different directions?
11. Table 1: What does this variable mean: overall survival (OS) or disease-free survival (DFS) or progression-free survival (PFS)?
12. Please explain in Table 1 column what exactly the three values (mean (minimum, maximum), median, etc.) indicate.
13. Please arrange the abbreviations in alphabetical order.
14. Line 293: There is no need to explain the chemical symbols. Please delete them.
15. Lines 331-340, Figure 6: If these images were generated with the software you used to create primers, please indicate this both in the figure legend and in the text.
16. Lines 364-365: As I mentioned at point 12) this information should be included in the table legend; if indeed the values from the above paragraph represent IQR, this is not very clear either in the text or in the table legend.
17. Line 366: Specify the level of statistical significance: for example, p = 0.05?
2. Results
18. Line 134, Figure 3: The labels of the axes should appear next to the axis, respectively under the X-axis, and the vertical panel with the ordinate. Fix Figure 3 accordingly to this format (refer to the PDF file attached).
19. Lines 112-113, 125-126, 138-139 and 141: I think the section "Results" should not be subdivided into subsections. These subsections represented in fact as paragraphs of text, and not as Chapter 2.3, represented after 2 lines. Please delete them or integrate them as a start phrase for each paragraph.
3. Discussions
20. Lines 156-264: Instead, I believe the "Discussion" section should be divided into 2 subsections: one in which you comment on the results of leptin isoforms’ roles in neoplasic proliferation as well as clinical implications of this using leptin isoforms of biomarkers for various purposes (this being a major part of the discussion), and another in which you discuss the role of information in the physiopathology and proliferation including mechanisms and signaling pathways, as you have already done, but unfortunately without presenting contrastively these pathways: which of them “fit better” to your results?
21. Is there any in the literature other citations related to other signaling pathways for tumor cell proliferation in adrenocortical neoplasms?
22. Lines 220-224, 260-264: The text is scientific and should be elaborated more in the conclusions.
5. Conclusions
23. Lines 369 -388: The conclusions are totally inappropriate, being a succession of generalities that seem to be generated automatically or at least compiled from different sources. Authors should in the conclusions summarize their main results, as well as the leptin’s revelator role on the mechanisms of oncogenesis and their clinical implications in diagnosis, monitoring, and prognosis of the disease.
References
24. Lines 410-537: The format of the cited works is incorrect. Some works are cited by the enumeration of all authors, and others are cited by mentioning the first author et al., which is not transparent. The list should be exhaustive and facilitate the verification of auto-citations and their relevance works.

No further comments on the quality of the English used in the manuscript.
Author Response
Abstract
- Line 20: I don't think it's very correct or advisable for a sentence to start with a number. Please modify it.
Thank you for pointing this out. We have modified this sentence.
- Lines 29-30: I think these conclusions should be formulated more specifically, especially since the authors obtained results that are quite clear. LepR1 can be a diagnostic marker for carcinomas, LepR3, and LepR4 can be used as prognostic markers.
Thank you for your suggestions. We have revised the Conclusions according to your recommendations (lines 29-32).
- Line 25: Instead of using the complicated notation "LepRv1" the authors might use simply "LepR1" following the notation model of the other receptor variants.
Agree. To address this, we have unified the nomenclature throughout the manuscript.
- Introduction
- Lines 44-48: See observation 3) above.
Agree. To address this, we have unified the nomenclature throughout the manuscript.
- Lines 60-65: The authors do not use any citations in this text. Indeed, the text contains textbook information but 1-2 citations would be welcome, especially for readers who are not very familiar with the subject and are doing preliminary research.
Thank you for your feedback. We have addressed your comment by adding a citation to the relevant sections of the text (https://doi.org/10.1098%2Frsob.210120).
- Line 79: Before listing the incidence and some clinical and pathophysiological characteristics of these tumors, a brief listing or even a brief classification of these would be useful. The authors should specify which pathological entities they will focus on in their investigation.
Thank you for your valuable suggestion. We believe we had included a brief classification of adrenal tumors in lines. We added information that our study specifically focuses on adrenocortical carcinoma, adenoma, hyperplasia, myelolipoma, and pheochromocytoma (lines 97-98).
- Lines 95-96: Use the same notation for the expression of the Ki67 protein (either Ki67 or Ki-67) throughout the manuscript.
We agree with this comment. To address this, we have unified the nomenclature throughout the manuscript.
- Line 108: The term "cancer" is quite imprecise, maybe the term "tumor" should be used (possibly enumerate).
Thank you for your observation. We understand the importance of precise terminology. However, we believe that the term "the NCI-H295R adrenocortical cancer cell line model" is accurate and appropriate in this context, as it refers to the specific name of the cell line used in our study (line 131).
- Materials and Methods:
- Line 265: A subsection, "Study Cohort and Strategy" (or something similar) for the text between the labels on lines 265-284 needs to be introduced considering there are major sections further below and also since this section includes the study group, the study stages, and the data collection methodology
Agree. We have added a subsection titled "Study Cohort and Strategy" (line 303)
- Line 289, Table 1: What values for this variable were recorded? Maximum values or an average, or were multiple diameters measured in different directions?
The table presents the median along with the 1st and 3rd quartile values. Relevant information is provided in the table description.
- Table 1: What does this variable mean: overall survival (OS) or disease-free survival (DFS) or progression-free survival (PFS)?
Overall survival. DFS, and PFS were excluded from the table and are not presented there.
- Please explain in Table 1 column what exactly the three values (mean (minimum, maximum), median, etc.) indicate.
Thank you for this point. Indeed, we did not include the meaning of the presented values in the table. In the current version, the table presents the median along with the 1st and 3rd quartile values. Relevant information is provided in the table description.
- Please arrange the abbreviations in alphabetical order.
Following the point 14. we have deleted the abbreviations.
- Line 293: There is no need to explain the chemical symbols. Please delete them.
Thank you for your comment. We have deleted the abbreviations.
- Lines 331-340, Figure 6: If these images were generated with the software you used to create primers, please indicate this both in the figure legend and in the text.
We added relevant information to the figure legend. The overall structure of the leptin receptor gene isoforms, with consideration of the location of individual exons, was analyzed using DNAsis Max ver. 2.5 (Hitachi Ltd., Tokyo, Japan). The primer binding site positions were marked manually using Adobe Illustrator ver. 28.4 (Adobe Inc., San Jose, CA, USA).
- Lines 364-365: As I mentioned at point 12) this information should be included in the table legend; if indeed the values from the above paragraph represent IQR, this is not very clear either in the text or in the table legend.
We added relevant information to the table legend.
- Line 366: Specify the level of statistical significance: for example, p = 0.05?
Relevant information was added to the text (line 408)
- Results
- Line 134, Figure 3: The labels of the axes should appear next to the axis, respectively under the X-axis, and the vertical panel with the ordinate. Fix Figure 3 accordingly to this format (refer to the PDF file attached).
The figure has been changed in accordance with the reviewer's suggestions
- Lines 112-113, 125-126, 138-139 and 141: I think the section "Results" should not be subdivided into subsections. These subsections represented in fact as paragraphs of text, and not as Chapter 2.3, represented after 2 lines. Please delete them or integrate them as a start phrase for each paragraph.
Thank you for your feedback. The subdivision of the "Results" section into subsections was based on the template suggested by the journal. However, we understand your concern and appreciate your input. We will consider this feedback and adjust accordingly in future revisions if needed.
- Discussions
- Lines 156-264: Instead, I believe the "Discussion" section should be divided into 2 subsections: one in which you comment on the results of leptin isoforms’ roles in neoplasic proliferation as well as clinical implications of this using leptin isoforms of biomarkers for various purposes (this being a major part of the discussion), and another in which you discuss the role of information in the physiopathology and proliferation including mechanisms and signaling pathways, as you have already done, but unfortunately without presenting contrastively these pathways: which of them “fit better” to your results?
Thank you for your insightful suggestions. We have reorganized the "Discussion" section according to your recommendations. Also we sugested that it is possible that leptin acting through LepR1 via MAPK- dependent signaling pathway inhibits cell proliferation in adrenal tumors (lines 286-287).
- Is there any in the literature other citations related to other signaling pathways for tumor cell proliferation in adrenocortical neoplasms?
Indeed, there are several recent studies in the literature that explore various signaling pathways involved in tumor cell proliferation in adrenocortical neoplasms, particularly in adrenocortical carcinoma (ACC). We have included a discussion of these additional pathways in the revised manuscript (lines 273-277).
- Lines 220-224, 260-264: The text is scientific and should be elaborated more in the conclusions.
Thank you for your feedback. We have revised the Conclusions section according to your suggestion.
- Conclusions
- Lines 369 -388: The conclusions are totally inappropriate, being a succession of generalities that seem to be generated automatically or at least compiled from different sources. Authors should in the conclusions summarize their main results, as well as the leptin’s revelator role on the mechanisms of oncogenesis and their clinical implications in diagnosis, monitoring, and prognosis of the disease.
Thank you for your feedback. We have revised the Conclusions section according to your suggestion (lines411-424).
References
- Lines 410-537: The format of the cited works is incorrect. Some works are cited by the enumeration of all authors, and others are cited by mentioning the first author et al., which is not transparent. The list should be exhaustive and facilitate the verification of auto-citations and their relevance works.
Thank you for your valuable feedback. We have carefully reviewed and corrected the references in our manuscript. As per the citation style guidelines, we have used "et al." when there are more than six authors for a work. We hope this addresses your concern and improves the transparency and accuracy of our references.
We are grateful for your numerous accurate and meticulous comments. Implementing your suggestions has significantly improved the quality of our publication. Thank you for your valuable contributions.

Round 2
Reviewer 1 Report
Comments and Suggestions for Authors
The authors have adequately taken into consideration most of my concerns.